# Routing Schemes in FANETs: A Survey

**DOI:** 10.3390/s20010038

**Published:** 2019-12-19

**Authors:** Muhammad Fahad Khan, Kok-Lim Alvin Yau, Rafidah Md Noor, Muhammad Ali Imran

**Affiliations:** 1Department of Computing and Information Systems, School of Science and Technology, Sunway University, Bandar Sunway 47500, Malaysia; 2Department of Computer Science, COMSATS University Islamabad (CUI), Attock Campus, Punjab 43600, Pakistan; 3Department of Computer System and Technology, Faculty of Computer Science and Information Technology, University of Malaya, Kuala Lumpur 50603, Malaysia; fidah@um.edu.my; 4School of Engineering, University of Glasgow, Glasgow G12 8QQ, UK; muhammad.imran@glasgow.ac.uk

**Keywords:** ad hoc networks, FANETs, routing, network topology

## Abstract

Flying ad hoc network (FANET) is a self-organizing wireless network that enables inexpensive, flexible, and easy-to-deploy flying nodes, such as unmanned aerial vehicles (UAVs), to communicate among themselves in the absence of fixed network infrastructure. FANET is one of the emerging networks that has an extensive range of next-generation applications. Hence, FANET plays a significant role in achieving application-based goals. Routing enables the flying nodes to collaborate and coordinate among themselves and to establish routes to radio access infrastructure, particularly FANET base station (BS). With a longer route lifetime, the effects of link disconnections and network partitions reduce. Routing must cater to two main characteristics of FANETs that reduce the route lifetime. Firstly, the collaboration nature requires the flying nodes to exchange messages and to coordinate among themselves, causing high energy consumption. Secondly, the mobility pattern of the flying nodes is highly dynamic in a three-dimensional space and they may be spaced far apart, causing link disconnection. In this paper, we present a comprehensive survey of the limited research work of routing schemes in FANETs. Different aspects, including objectives, challenges, routing metrics, characteristics, and performance measures, are covered. Furthermore, we present open issues.

## 1. Introduction

Flying ad hoc network (FANET) is a branch of networking that provides communication among flying nodes, particularly unmanned aerial vehicles (UAVs), with base station (BS) [1]. Flying UAVs are autonomous nodes capable of making decisions (e.g., changing speed and direction) in the air in a distributed manner rather than receiving decisions made from the ground in a centralized manner [2,3]. FANET has distinguishing features compared to existing ad hoc networks such as mobile ad hoc networks (MANETs) [4] and vehicular ad hoc networks (VANETs) [5]. In general, the flying nodes are prone to frequent link disconnections and network partitions due to the following:
Movement in the three-dimensional space (against movement that is confined to roads and highways in VANETs [6]).Flying speed ranges from 30 to 460 km/h [7] (against driving speed that ranges from 10 to 120 km/h [8,9]).Low node density (against high node density in urban areas in VANETs). In addition, the flying nodes, which are generally equipped with batteries that provide limited energy, have high energy consumption [10].

This means that frequent disconnections cannot be addressed by increasing the transmission power to provide long-range communication in FANETs. Hence, establishing long-term, reliable, and robust connections and routes is essential to increase route lifetime and to improve quality of service (QoS) (e.g., low latency and route setup time), yet it is challenging in FANETs. Nevertheless, the flying nodes naturally have a) a large coverage and b) minimal effects of obstacles (e.g., buildings and trees) and bad weather due to its elevated look angle [7].

Figure 1 shows two scenarios, namely the single-UAV and multi-UAV scenarios. In the single-UAV scenario, a single UAV establishes a connection to a radio access infrastructure (i.e., FANET base station (BS), cellular BS, and satellite). In the multi-UAV scenario, multiple UAVs cooperate and establish an ad hoc network. Compared to the single-UAV scenario, the multi-UAV scenario provides three main advantages [11]. Firstly, FANETs increase network scalability as multiple UAVs can increase coverage [12,13,14]. Secondly, FANETs increase network survivability (or network robustness and reliability) in a dynamic operating environment (e.g., due to poor weather condition) as multiple UAVs can either connect with each other to form an ad hoc network or to connect to radio access infrastructure directly [15]. The presence of a large swarm of UAVs (or multi-UAV swarm) [16,17] can cater to the failure of a UAV during operations [18,19]. Thirdly, FANETs can distribute payload among UAVs, which helps to reduce the weight of each UAV and, hence, the required energy for the UAV to reach and maintain a targeted altitude, leading to a longer route lifetime [20].

The multi-UAV scenario can be extended to multi-UAV swarm, whereby a large swarm of autonomous, small-sized, and lightweight UAVs are deployed. The multi-UAV swarm can coordinate and prevent collisions among themselves while completing tasks that require a large swarm of UAVs, such as surveillance [16] and search and rescue missions during catastrophe [17] and gathering a swarm of UAVs at a target location [21]. The multi-UAV swarm can be used to upload a huge amount of data collected in a distributed manner (e.g., data collected from cellular users) to the base station. In Reference [22], a multi-UAV swarm with high mobility uses a store-carry-forward approach to capture and transfer images and videos from a post-disaster area to the base station. In Reference [23], a multi-UAV swarm senses and allocates subchannels for UAV-to-X communication in a collaborative manner in order to maximize the sum-rate of uplinks while optimizing the speed of UAVs. In Reference [24], a multi-UAV swarm, which consists of flexible aerial nodes, is deployed to form an emergency network that can recover communication rapidly in a catastrophic area. In Reference [25], UAVs embedded with solar panels, together with solar-powered charging stations, are deployed to overcome energy constraint in order to fulfill the energy, communication, and safety requirements of 5G. In Reference [26], Euclidean distance is measured and it is used to adjust the transmission power for hello messages in order to reduce energy consumption. The number of UAVs required to achieve network performance requirements, such as throughput and packet delivery ratio, are also considered with the objective of minimizing energy consumption. In Reference [27], the secrecy outage probability and the average secrecy capacity of a multi-UAV swarm are derived in order to prevent eavesdropping.

### 1.1. Our Contribution

In view of the limited research work of routing schemes in FANETs and the lack of focus on the main characteristics of FANETs, such as the three-dimensional movement and mobility, this paper provides a comprehensive survey to stimulate research interest in this topic. Various routing schemes are classified, analyzed, and discussed based on a taxonomy. Open issues are outlined.

Table 1 summarizes papers conducting surveys on FANETs, covering various foci, which are self-explanatory given examples, as follows:Motivation for the need of FANETs and an explanation of their unique characteristics;Comparison with other ad hoc networks, such as MANETs and VANETs;Requirements, such as the bandwidth and energy requirements of the flying nodes;Mobility models of the flying nodes, such as random waypoint and Manhattan;Taxonomy that covers various FANET attributes;Objectives, such as achieving the awareness of energy consumption and cost;Challenges, such as high energy consumption and number of retransmissions;Routing metrics, such as the residual energy level and the distance between flying nodes;Characteristics, such as the transmission range of flying nodes and the number of nodes in a network;Performance measures, such as the number of clusters in a network and cost;Open issues, such as enhancing QoS and addressing the dynamicity of network topology.

To the best of our knowledge, this paper is the first of its kind to provide a comprehensive survey of routing schemes in FANETs, covering a diverse range of aspects. As shown in Table 1, surveys of routing schemes in FANETs have been conducted in the literature with a diverse range of foci [18,32,35,37,39,41], such as hierarchical and data centric approaches [18]; geographical location-based approaches [35]; as well as proactive, reactive, and hybrid routing schemes based on MANETs and VANETs [32]. In addition, although existing papers (see Table 1), including the surveys of routing schemes in FANETs, present taxonomies that categorize routing schemes and mobility models in FANETs, this paper presents a taxonomy that covers various aspects, including objectives, challenges, routing metrics, characteristics, and performance measures.

### 1.2. Significance of Our Work

Apart from being a comprehensive survey that covers a diverse range of foci as shown in Table 1, this paper provides some distinguishing aspects. Firstly, it provides motivation for the topic, answering questions such as: “How do FANETs differ from other kinds of ad hoc networks?”, “What are the roles of FANETs?”, “What are the requirements of routing in FANETs?”, “What are the mobility models in FANETs?”, and “What are the roles of artificial intelligence in FANETs?”. Secondly, a comprehensive taxonomy for routing in FANETs is provided, and it is used to capture various aspects of the state-of-the-art routing schemes in FANETs. Finally, some significant open issues in this topic are first presented for the first time in the literature. Hence, this paper has laid a strong foundation for future investigations in routing in FANETs.

### 1.3. Organization of This Paper

The rest of this paper is organized as follows:Section 2 presents background and the motivation for the need to investigate routing schemes in FANETs.Section 3 presents the taxonomy and framework of routing schemes in FANETs.Section 4 presents the framework of routing schemes in FANETs.Section 5 presents a discussion of various routing schemes in FANETs based on the taxonomy.Section 6 presents open issues.Section 7 concludes the paper.

## 2. Background and Motivating the Need for FANETs

FANET has distinguishing features (e.g., high dynamicity of network topology, large coverage, and limited energy) as compared to other kinds of ad hoc networks (e.g., MANETs and VANETs). Hence, in FANETs, flying nodes can join and leave a highly dynamic ad hoc network, which causes frequent link disconnections and route breakages. This has motivated researchers to investigate FANETs.

### 2.1. How Do FANETs Differ from Other Kinds of Ad Hoc Networks?

Ad hoc network is composed of geographically distributed connected devices that can communicate with each other over a wireless medium. In general, it is deployed to support short-term applications, such as military usage, video conferencing, infotainment, as well as disaster relief and rescue operation, for a short time period. The ad hoc network is different from a cellular network because the former, which lacks a fixed infrastructure and powered by battery, must perform computation in a distributed manner; hence, energy efficiency is a major concern [42,43]. Table 2 compares and contrasts the three main kinds of ad hoc networks, namely MANETs, VANETs, and FANETs, shown in Figure 2. Due to the differences among the ad hoc networks, novel routing schemes must be designed to cater to the main characteristics of FANETs.

#### 2.1.1. How Do FANETs Differ from MANETs and VANETs in Terms of the Types of Links?

Table 2 compares the types of links in FANETs with those in MANETs and VANETs [44,45,46,47].

There are four main types of links among UAVs and different radio access infrastructure (i.e., FANET BS, cellular BS, and satellite) in ad hoc networks [48,49] as shown in Figure 3:a UAV–UAV link between a UAV and another UAV that supports ad hoc communication. This link is part of a route so that intermediate UAVs can forward packets towards a radio access infrastructure.a UAV–BS link (or a direct link) between a UAV and a FANET BS.a UAV–cellular link between a UAV and a cellular BS.a UAV–satellite link, which is a long-range terrestrial link, between a UAV and a satellite. The UAVs can form a star topology with the satellite. This link is necessary in the absence of other radio access infrastructure (i.e., FANET BS and cellular BS).

The radio access infrastructure is prone to failure during disaster. Hence, the ad hoc network, which is formed using UAV–UAV links, is more robust and easy to deploy, although routing is necessary to establish routes.

#### 2.1.2. How Do FANETs Differ from MANETs and VANETs in Terms of Characteristics?

Table 2 compares the parameters and characteristics of FANETs with those of MANETs and VANETs as follows:Mobility degree (or the dynamicity of network topology) of FANETs is high (i.e., 30–460 km/h [20]) compared to MANETs (i.e., 5–50 km/h [50]) and VANETs (i.e., 10–120 km/h [8,9]). Hence, in FANET, the highly dynamic network topology causes frequent link disconnections and network partitions, resulting in low link quality [51,52].Mobility models of FANETs is different in terms of the capabilities of devices (e.g., UAVs move in the three-dimensional space, while nodes and vehicles in MANETs and VANETs move in a two-dimensional space) and the operating environment (e.g., FANETs operate in the sky, while MANETs operate in the terrain most of the time using random waypoint as it also considered harmful [53] but other model in random trip model [54] and VANETs operate in the highways using prediction based model). Examples of mobility models for FANETs are semi-random circular movement (SRCM) model [55,56], realistic model, and random waypoint model [57]. More details about the mobility models are presented in Section 2.4.Energy constraint in FANETs is moderate [7] compared to MANETs (i.e., has the highest energy constraint) and VANETs (i.e., has the lowest energy constraint). The availability of energy affects route lifetime, and so, small UAVs must conserve energy to support long flight time. In VANETs, vehicles are generally powered by vehicular battery with less energy constraint.Availability of line of sight (LOS) between UAVs is commonplace in FANETs contributing to higher robustness of a link, while there may be non-LOS (NLOS) in MANETs and VANETs as a result of obstacles in a link.Localization method in FANETs, such as inertia measurement unit, provides accurate coordinates of a UAV and neighboring UAVs in a real-time manner [58]. In contrast, MANETs use GPS to find the coordinates of nodes with an accuracy of 10–15 m [59], while VANETs use assisted-GPS or differential-GPS with an accuracy of 10 cm to reduce collisions and find routes [60].Node density (or the number of nodes in a unit area) of FANETs is low compared to MANETs and VANETs. This means that UAVs are spaced far apart in the sky and that the distance between the UAVs is comparatively long. Lower node density can increase the effects of the dynamicity of network topology. Nevertheless, node density can be high for multi-UAV swarm whereby a large swarm of UAVs is deployed.

### 2.2. What Are the Roles of FANETs?

FANETs provide connectivity among flying nodes (e.g., UAVs) to support various short-term applications and tasks [61]. Multiple UAVs can be deployed to fly over a targeted region and to perform surveillance, detection, and monitoring. The UAVs use sensors and cameras to capture real-time images, audios, and videos at remote and difficult-to-access areas and send the data to a BS immediately [62]. Subsequently, the BS processes the data and generates messages, such as alert messages upon the detection of a disturbance or an event [63]. The multi-UAV scenario has a significant impact to human life and activities [64]. For instance, the use of the multi-UAV scenario to detect and monitor wildfire is foreseen to reduce the current 340,000 cases of casualties and the $10 million USD cost annually [62,65].

The multi-UAV scenario has been deployed to detect targets (i.e., to identify the position of a target) in search missions [66], to provide bird’s-eye view for surveillance [67,68,69], to monitor crops (e.g., identifying ripe and unripe crops) in agriculture [70,71], to monitor disaster [72,73,74], and to monitor environment (e.g., wind [75], temperature, humidity, light intensity, and the pollution level [76,77,78]). Multi-UAVs also support other kinds of ad hoc networks, particularly MANETs and VANETs. For instance, the multi-UAV scenario has been deployed to monitor and manage traffic in VANETs [79,80]. The multi-UAV swarm scenario has been deployed to monitor and manage disaster and to perform surveillance in smart cities [81].

### 2.3. What Are the Requirements of Routing in FANETs?

Routing in FANETs has four requirements as follows:High adaptability since UAVs must adapt to the highly dynamic network topology with low node density and cater to link disconnections and network partitions [7]. Hence, route discovery (i.e., establishing routes for data dissemination) and maintenance (i.e., reestablishing routes) must be sufficiently adaptive to improve route reliability (or robustness) in FANETs. This means that the routing table, which maintains the routes and their route costs, must be constantly updated and that reliable routes must be identified.High scalability since UAVs must cater to large-scale applications that require multiple UAVs [82] with high or low node density. Hence, route discovery and maintenance must be supported by collaboration and coordination among UAVs to improve network scalability.High residual energy since UAVs, which are powered by battery, must establish routes with sufficiently high residual energy to reduce link disconnections and network partitions as a result of node failures [83] in order to prolong route lifetime.Low latency since UAVs must cater to real-time (or delay-sensitive) applications, such as collision prevention in multi-UAV swarm, as well as disaster relief and rescue operation. Hence, route discovery and maintenance must reduce latency as investigated in References [84,85].High bandwidth since UAVs must gather data or sensing outcomes from a single or multiple locations and send them to radio access infrastructure for processing and decision making. Hence, route discovery and maintenance must establish routes with high (or sufficiently high) bandwidth.

### 2.4. What Are the Mobility Models in FANETs?

Mobility models have been used to characterize the movement of UAVs, including the speed, direction, and acceleration of UAVs, in a fixed space. Such models have been used to develop real-time simulation environment for FANETs. The rest of this section presents various kinds of mobility models used in the investigation of FANETs.

#### 2.4.1. Pure Randomized Mobility Models

The pure randomized mobility model has been widely used in ad hoc networks. There are four common characteristics. Firstly, nodes are independent on each other. Secondly, nodes are memoryless, and so, the new speed and direction of the nodes are independent on their previous speed and direction [86]. Thirdly, nodes can move at any speed within a range (e.g., 30–460 km/h [20]). Fourthly, nodes can move freely in an unpredictable manner in a fixed space at all times. The difference among the pure randomized mobility models lies in the way in which the direction of the nodes’ movement is determined.
A.1Random walk, which is based on the Brownian movement, allows a node to move in any directionsA.2Random waypoint [87,88] allows a node to move in different directions (i.e., either straight, left, or right, rather than backward) towards a destination [89,90]. When the node arrives at its destination inside a fixed space, it stops for a short period of time, which helps to prevent a drastic change in order to provide smooth movement, and then, it starts to move towards another randomly chosen destination.A.3Random direction (or random mobility) shares the similar way in which the direction of a node’s movement is determined with random waypoint (A.2). The main difference is that a node must stop at the edge of a fixed space [91]. This means that a destination must be located at the edge of a fixed space rather than inside the space.A.4Manhattan grid allows a node to move in straight *x*-axis and *y*-axis directions in a grid road layout in urban areas. Specifically, the node moves in different directions (i.e., either straight, left, or right, rather than backward) to proceed or to turn to the left or right road at an intersection [92].

#### 2.4.2. Path-Planned Mobility Models

The path-planned mobility model forms predefined routes used by UAVs. There are three common characteristics. Firstly, nodes are independent on each other. Secondly, nodes are memoryless, and so, the new speed and direction of the nodes are independent on their previous speed and direction. Thirdly, nodes can move at any speed within a range.
A.5Semi-random circular movement (SRCM) allows a node to form a circular (or curve) route [55]. When the node arrives at its destination following the route, it stops for a short period of time, and then, it starts to move towards another randomly chosen destination using a newly predefined route. The circular routes have been shown to reduce collisions between UAVs. This model is suitable for multi-UAV scenario.

#### 2.4.3. Time-Dependent Mobility Models

The time-dependent mobility model minimizes acceleration and the drastic changes of direction in order to provide smooth movement (e.g., a smooth turn), which is more realistic [93]. There are three common characteristics of this model. Firstly, nodes are dependent on each other, whereby the vehicular nodes follow one another (including speed and direction) on the road. Secondly, nodes are memory-based, and so, the new speed and direction of the nodes are dependent on their previous speed and direction. Thirdly, nodes can move at any speed within a range.
A.6Smooth turn (or realistic model) allows a node to choose a point and to form a circle around it, and then, it moves along the circle [55]. When the node moves towards another randomly chosen destination, it chooses a new point and moves along another circle around the newly chosen point. Nevertheless, there is lack of collision avoidance [94], which explain the need for better collaboration among nodes. This model is suitable for multi-UAV scenario.

### 2.5. What Are the Roles of Artificial Intelligence in FANETs?

Artificial intelligence (AI) approaches, including artificial neural network [95], fuzzy logic [96], reinforcement learning [97], and particle swarm optimization [98], has been adopted to improve the performance of complex systems in a diverse range of applications, such as FANETs. In FANETs, AI has been applied in routing [99] so that a decision maker observes the state of the operating environment, selects the best possible action (e.g., a stable and reliable route), receives reward, and learns about the appropriateness of the action under the state based on the reward. For instance, a flying node (decision maker) observes the mobility pattern of neighboring nodes (state), selects a stable and reliable route (action), receives throughput performance (reward), and learns the best possible route as time goes by [100]. In Reference [99], AI is used to predict the geographical location of flying nodes in the selection of next-hop node in FANETs. In Reference [101], AI is used to assist the nodes in VANETs and MANETs. In Reference [102], AI is used to form clusters and improve energy efficiency. In Reference [103], a swarm-based approach is applied to routing in a multi-UAV swarm scenario.

## 3. Taxonomy

This section presents a taxonomy of routing attributes in FANETs as shown in Figure 4. The rest of this subsection explains the taxonomy.

### 3.1. Objectives

There are three main routing objectives in FANETs:O.1Enhancing route stability. In FANETs, route stability and route lifetime are lower because a) the network topology is highly dynamic due to high mobility degree and b) the distance between nodes in FANETs is comparatively long. Hence, frequent route discovery and maintenance are needed to reestablish routes. In addition, route stability increases with node density and residual energy.O.2Enhancing network coverage. In FANETs, node density is lower, so a large transmission range and network coverage is needed to reduce the effects of link disconnections and network partitions in order to improve network connectivity. In addition, network coverage increases with node density in multi-UAV scenario.O.3Enhancing routing performance and QoS. Since route stability is lower in FANETs, the routing performance reduces due to increased route discovery and maintenance. Hence, suitable routes (e.g., routes that provide high route stability and route lifetime, and routes with lesser number of hops that reduce the latency of data dissemination in an ad hoc manner) are identified and selected. In addition, QoS increases with node density and transmission range, which increases network connectivity.

In general, route stability is higher, network coverage is higher, and routing performance improves:higher QoS (e.g., higher throughput, as well as lower latency and packet loss);lower routing cost;longer route lifetime;lower latency of data dissemination in an ad hoc manner.

### 3.2. Challenges

There are three main challenges that must be addressed in routing for FANETs in order to improve routing performance, such as higher QoS performance and route setup success rate, as well as lower energy consumption and number of clusters (see Section 3.5):X.1High dynamicity. In FANETs, the highly dynamic network topology, as a result of high mobility, causes low link quality. Consequently, there are frequent link disconnections and network partitions, which increase route discovery and maintenance, resulting in lower routing performance. Various mobility models (see Section 2.4) of UAVs has been designed to investigate routing [104].X.2High cost. In FANETs, frequent route updates, route discovery (or to reestablish routes), route maintenance (as well as packet retransmission) due to route failure, can incur three main types of costs: (a) routing overhead (or inefficient resource utilization); (b) energy consumption; and (c) computational cost due to the time incurred for route discovery and maintenance. Multi-UAVs can be deployed to increase connectivity among UAV [7].X.3Low residual energy. In FANETs, UAVs, which are powered by battery, have limited energy (a) to perform routing (i.e., route updates, discovery, and maintenance); (b) to retransmit packets due to route failure; and (c) to support long transmission range due to the comparatively long distance between UAVs. Meanwhile, UAVs with higher payloads increase energy consumption [7].

### 3.3. Routing Metrics

There are five main routing metrics used for selecting routes in FANETs:M.1Mobility metrics, such as speed, distance, and angle of arrival. This metric helps to achieve two main objectives in FANETs, namely enhancing route stability (O.1) and enhancing routing performance and QoS (O.3) because nodes with similar speed or angle of arrival are selected as part of a route to prolong the route lifetime.M.2Link expiration time, which depends on the distance between two UAVs. Longer link expiration time increases the link lifetime and, hence, the route lifetime. This metric helps to achieve two main objectives in FANETs, namely enhancing route stability (O.1) and enhancing routing performance and QoS (O.3) because (a) nodes with a higher number of neighbor nodes, which tend to have higher link expiration time, are selected as part of a route to increase route stability and (b) a new route can be reestablished before a link expires based on the link expiration time to ensure smooth network operation.M.3Geographical location can be obtained from GPS. In general, nodes are spaced far apart, and so, physically closer neighbor nodes are selected as part of a route; however, this may increase the number of hops of a route. This metric helps to achieve two main objectives in FANETs, namely enhancing route stability (O.1) and enhancing routing performance and QoS (O.3) because physically closer neighbor nodes can increase route lifetime and reduce route updates, discovery, and maintenance.M.4Residual energy can be used to reduce routing overhead incurred in route discovery and maintenance caused by link disconnections and network partitions, contributing to lower energy consumption. This metric helps to achieve three main objectives, namely enhancing route stability (O.1), enhancing network coverage (O.2), and enhancing routing performance and QoS (O.3) (e.g., higher throughput with reduced route discovery and maintenance) because neighbor nodes with higher residual energy are selected as part of a route to improve network connectivity among UAVs.M.5Node identity, such as node ID, can be used to select intermediate nodes, which receives packets and forwards them towards the destination, as part of a route in a random manner. This metric can be used along with the aforementioned routing metrics to achieve routing objectives.

### 3.4. Characteristics

Routes in FANETs can be characterized by the following:C.1Routing decision can be made at either the source node or the intermediate node. There are two options as follows:
C.1.1Hop-by-hop routing includes the next-hop node and destination node information (i.e., node addresses) in the route request (RREQ) message. Hence, routing tables must be maintained at intermediate nodes to store the information [105].C.1.2Source routing includes the complete route information (i.e., the node addresses of all nodes of a route) in the RREQ message. Hence, routing table is not embedded in intermediate nodes, and these nodes can forward packets to destination nodes using the complete route information in RREQ.

### 3.5. Performance Measures

There are four main performance measures for routing in FANETs.
P.1Higher QoS performance increases throughput and packet delivery rate (or reduces packet loss rate) as well as reduces end-to-end delay of delay-sensitive packets.P.2Lower energy consumption increases route lifetime.P.3Lower number of clusters increases the cluster size and, hence, the coverage of each cluster, which is a logical group of nodes comprised of the leader of a cluster (or cluster head, CH) and cluster members (CMs). Larger cluster size reduces intra-cluster communication (i.e., between CH and CMs from the same cluster) and inter-cluster communication (i.e., between CH and CMs from different clusters) [106,107].P.4Higher route setup success rate increases with the capability of setting up a route between a source node and a destination node.

## 4. Routing Framework

Routing framework consists of steps for nodes to find routes from source to destination nodes. In general, there are three main steps. Firstly, neighbor and route discoveries enable nodes (a) to broadcast control messages (e.g., hello messages) periodically to identify neighboring nodes in order to form an up-to-date neighbor set as the network topology changes as time goes by and (b) to exchange routing messages, whereby the source node sends messages, such as route request (RREQ), to the destination node to gather node information (e.g., node addresses and geographical location) along potential routes and the destination node sends messages, such as route reply (RREP), to the source node using the same route that the RREQ message traverses. Secondly, route selection enables nodes to use routing metrics to select routes. For instance, link expiration time (M.2) can be used to prolong route lifetime. Thirdly, route maintenance enables nodes to reestablish routes in order to cater for the dynamicity of network topology. There are four main categories of routing frameworks based on how route selection is made in FANETs as shown in Table 3. The rest of this section presents the frameworks of the routing schemes.
F.1Adaptive: The adaptive routing scheme learns, selects, and changes routes dynamically based on the current network condition (e.g., network congestion level and network connectivity) and the updates of the routes. This enables nodes to select the optimal routes based on knowledge as time goes by.The advantage is that packets can be sent along optimal routes, contributing to reduced link disconnections and network congestion and, hence, improved network performance. Nevertheless, there is a shortcoming whereby computational complexity increases due to learning.Some adaptive routing schemes used in MANETs, VANETs, and FANETs are adaptive routing in dynamic ad hoc networks (AROD) [108], static-node assisted adaptive routing (SADV) [109], and self-learning routing protocol based on reinforcement learning (RLSRP) [100].F.2Proactive: The proactive (or active) routing scheme determines routes prior to packet arrivals to provide immediate data transmission; in other words, routes leading to all destination nodes in the network are established, stored, and maintained in routing tables in advance at UAVs. Since a route leading to a destination node is either readily available or unavailable at a source node, packets can be either sent or dropped immediately.The advantage is that packets can be sent immediately without any initial delay incurred to establish routes. Nevertheless, there are two shortcomings. Firstly, nodes must exchange a large number of messages (or routing overheads) among themselves periodically to establish, update, and maintain routes in the routing table due to the high dynamicity of network topology in which link disconnections and network partitions are commonplace. Consequently, routing overhead is high and bandwidth utilization is inefficient. Secondly, routes in the routing table may not be responsive to the dynamicity of network topology, and so, packets are sent along nonoptimal routes.Some proactive routing schemes used in MANETs, VANETs, and FANETs are optimized link state routing (OLSR) [110], destination-sequenced distance vector [111], and directional optimized link state routing protocol (DOLSR) [112].F.3Reactive: The reactive routing scheme determines routes upon packet arrivals in an on-demand manner; in other words, routes are established whenever data transmission is needed.The advantage is that it addresses the high routing overhead issue found in proactive routing. Specifically, it reduces the periodic exchange of routing messages among nodes and the computational cost incurred to maintain and reestablish routes periodically, contributing to higher efficiency of bandwidth utilization. Nevertheless, there is a shortcoming that the initial delay incurred to establish routes increases.Some reactive routing schemes used in MANETs, VANETs, and FANETs are ad hoc on-demand distance vector (AODV) [113], time-slotted AODV [114], modified-AODV [115], dynamic source routing (DSR) [116], and UAV-assisted vehicular network routing (UVAR) [101].F.4Hybrid: The hybrid routing scheme integrates both proactive and reactive routing schemes. In Reference [117], nodes in the networks are segregated into clusters. The intra-cluster communication (i.e., between CH and CMs from the same cluster) is supported by reactive routing, and the inter-cluster communication (i.e., between CH and CMs from different clusters) is supported by proactive routing. Hybrid routing scheme has been proposed to cluster multi-UAV swarm and select optimal routes over a clustered network [118].The advantage is that it reduces routing overheads from source to destination nodes as intra-cluster communication can be excluded in route discovery.Some hybrid routing schemes used in MANETs, VANETs, and FANETs are ant colony optimization [119], grey wolf optimization [120,121], moth flame optimization [122,123], and energy aware link based clustering [102].

## 5. Routing Schemes in FANETs

This section presents a survey on routing schemes in FANETs, and a summary of the routing schemes is shown in Table 4. Based on the frameworks presented in Section 4, the routing schemes are segregated into four categories, namely adaptive, proactive, reactive, and hybrid routing schemes.

### 5.1. Adaptive

This section presents two adaptive routing schemes for FANETs.

#### 5.1.1. Enhancing Routing Performance Using Reinforcement Learning

Zheng et al. [100] propose an adaptive routing scheme that enables each node to use reinforcement learning (RL), which is an artificial intelligence approach, to learn from its operating environment and to select the most suitable next-hop node as part of a route in FANETs. The main objectives are to enhance route stability (O.1) as well as to enhance routing performance and QoS (O.3) in FANETs. The routing scheme addresses two challenges, namely high dynamicity (X.1) of network topology as flying nodes have high mobility and cost (X.2), particularly energy consumption and computational cost. The routing scheme is designed for hop-by-hop routing (C.1.1), specifically a next-hop node is selected as part of a route. The underlying medium access control (MAC) has three main characteristics to support routing. Firstly, nodes exchange handshaking messages, such as ready-to-send, wait-to-send, and clear-to-send, to clear a channel in order to increase link reliability. Secondly, there are two transceivers used for concurrent transmission and reception of control messages and data packets, respectively. Examples of information in control messages are geographical locations and handshaking messages. Thirdly, directional antennas are used, whereby the antennas are directed properly (i.e., the antennas are directed towards the transmitter at the receiver and towards the receiver at the transmitter) after a successful handshake, which helps to improve the link stability and, hence, the route stability.

During the neighbor and route discoveries stage, each node sends control messages that contain mobility metrics (M.1) (i.e., nodes’ velocity), node identity (M.5) (i.e., node addresses), and the geographical locations (M.3) (i.e., GPS coordinates) of one-hop neighboring nodes to form a neighbor set comprised of two-hop neighboring nodes as well as to predict the next geographical location of the neighboring nodes based on their velocity. During the route selection stage, each node selects a next-hop node, which is in the idle state after a successful handshake, that provides the shortest route with highest possible packet deliver rate towards the destination node. RL is embedded in each node, and it has three main representations, including the following:state represents the status of the node (i.e., whether ready-to-send, wait-to-send, or clear-to-send in a handshake);action represents whether to transmit data packets;reward represents positive reward (i.e., when packets arrive at the destination node) or negative reward (i.e., when packets fail to arrive at the destination node).

RL enables each node to observe its state, to learn, and to take the right action at the right time in order to maximize its reward as time goes by. During the route maintenance stage, which is required when the accumulated reward of the node becomes lower than a predefined threshold as a result of packet loss, the node initiates the route discovery mechanism (or the first stage).

The routing scheme is compared with a state-of-the-art scheme that utilizes directional antennas and location estimation within the MAC layer, which uses unicasting and geo-casting routing approaches based on location and trajectory information so as to keep track of high-speed flying UAVs of which the topology changes. Random waypoint mobility model (A.2) is used in simulation. The routing scheme has been shown to increase QoS (P.1) (i.e., lower delay), route setup success rate (P.4) (i.e., packet deliver rate), and route lifetime.

#### 5.1.2. Enhancing Routing Performance Using Clustering

Khelifi et al. [99] propose an adaptive routing scheme that enables each node to use fuzzy logic to predict the next geographical location of the neighboring nodes based on their received signal strength in scenarios with unknown geographical locations (e.g., the GPS does not function) and select the most suitable next-hop node as part of a route in FANETs. The main objective is to enhance routing performance and QoS (O.3) in FANETs. The routing scheme addresses three challenges, namely high dynamicity (X.1) of network topology, cost (X.2), as well as low residual energy (X.3) as a result of high energy consumption and computational cost since flying nodes are spaced far apart in the sky. The routing scheme is designed for both hop-by-hop (C.1.1) and source (C.1.2) routing approaches.

During the neighbor and route discoveries stage, each node sends control messages that contain node identity (M.5) (i.e., node addresses) to form a neighbor set comprised of two-hop neighboring nodes as well as to predict the geographical locations of one-hop neighboring nodes based on received signal strength, which indicates the distance between nodes, using fuzzy logic [127]. During the route selection stage, in order to reduce energy consumption, the underlying network is clustered based on a) residual energy (M.4), whereby nodes with higher residual energy are selected as CHs, and b) geographical location (M.3), whereby nodes near a CH joins its cluster. The CH collects data from CMs and then aggregates and forwards the data. Selecting CHs with high residual energy and node density reduces energy consumption and, hence, prolongs route lifetime. During the route maintenance stage, which is required when neighboring nodes move out of a node’s transmission range, the node initiates the route discovery mechanism (or the first stage).

The routing scheme is compared with a state-of-the-art scheme which is an energy-efficient multi-hop hierarchical routing protocol that optimizes clustering process based on an energy-efficient strategy. Random direction mobility model (A.3) is used in simulation. The routing scheme has been shown to increase QoS (P.1) (i.e., higher throughput) and to reduce energy consumption (P.2).

#### 5.1.3. Enhancing Routing Performance Using Swarm-Based Clustering

Ali et al. [128] propose an adaptive routing scheme that minimizes the issue of unstable routes caused by limited battery and high mobility of UAVs. The proposed scheme uses the glowworm swarm optimization algorithm in an energy-aware cluster formation, which includes CH election. The main objective is to enhance route stability (O.1) as well as to enhance routing performance and QoS (O.3) in FANETs. The routing scheme addresses three challenges, namely high dynamicity (X.1) of network topology and cost (X.2) as well as low residual energy (X.3) as a result of high energy consumption and computational cost since flying nodes are spaced far apart in the sky. The routing scheme is designed for both hop-by-hop (C.1.1) and source (C.1.2) routing approaches.

During the neighbor and route discoveries stage, each node sends control messages that contain residual energy (M.4). Each node forms a neighbor set, and the neighbors are sorted according to a fitness value, which is calculated based on their respective residual energy levels. During the route selection stage, in order to reduce energy consumption, the underlying network is clustered based on mobility metrics (M.1), whereby nodes join a cluster in which its speed is near the speed of the CH of the cluster. The CH collects data from CMs and then aggregates and forwards the data. Therefore, selecting CHs with higher residual energy level and node density reduces energy consumption, and hence, it prolongs route lifetime. For route selection, a route is selected based on geographical location (M.3) and residual energy (M.4) of nodes. During the route maintenance stage, which is required when the residual energy level of CHs or CMs reaches the minimum threshold, the CHs and CMs initiate the route discovery mechanism, which is the first stage.

The routing scheme is compared with a state-of-the-art, which is a swarm-based routing scheme that provides an optimized number of clusters for routing. Random direction mobility model (A.3) is used in simulation. The routing scheme has been shown to increase QoS (P.1) (i.e., lower cluster formation time) and route setup success rate (P.4) as well as to reduce energy consumption (P.2).

### 5.2. Proactive

This section presents three proactive routing schemes for FANETs.

#### 5.2.1. Enhancing Routing Performance Using Shortest Route

Alshabtat et al. [112] propose a proactive routing scheme for nodes equipped with directional antennas to find the shortest route, which has the least possible number of nodes in a route while reducing routing overhead. The main objectives are to enhance network coverage (O.2) as well as to enhance routing performance and QoS (O.3), in FANETs. The routing scheme addresses two challenges, namely high dynamicity (X.1) of network topology and cost (X.2), particularly routing overhead. The routing scheme is designed for hop-by-hop routing (C.1.1) approach.

During the neighbor and route discoveries stage, each node broadcasts hello messages to one-hop neighboring nodes to form a neighbor set comprised of two-hop neighboring nodes. The hello messages are not forwarded to reduce cost, particularly routing overhead and energy consumption. During the route selection stage, each node selects a next-hop node by using mobility metrics (M.1) (i.e., the relative speed of nodes), whereby neighbor nodes with similar mobility metrics are selected. The farthest node is selected in case of more than a single potential next-hop node in order to reduce the number of hops of a route, which helps to reduce the end-to-end delay of packet transmission. During the route maintenance stage, link disconnections are detected by the absence of the periodic hello messages. The intermediate node sends a link failure notification message to the source node in order to initiate a route discovery mechanism [110].

The routing scheme is compared with a state-of-the-art scheme that aims to provide lower end-to-end delay. Random waypoint (A.2) and semi-random circular movement (A.5) mobility models are used in simulation. The routing scheme has been shown to increase QoS (P.1) (i.e., higher throughput, as well as lower end-to-end delay and routing overhead).

#### 5.2.2. Enhancing Route Stability Based on Link Connectivity

Rosati et al. [124] propose a proactive routing scheme that enables each node to select a next-hop node with the maximum link connectivity time in order to extend coverage in FANETs. The main objectives are to enhance route stability (O.1) as well as to enhance routing performance and QoS (O.3) in FANETs. The routing scheme addresses the challenge of high dynamicity (X.1) of network topology, which causes frequent updates of routing table. This routing protocol is designed for source routing (C.1.2).

During the neighbor and route discoveries stage, each node broadcasts hello messages, which contains the mobility metrics (M.1) (i.e., nodes’ velocity), node identity (M.5) (i.e., node addresses), and the geographical locations (M.3) (i.e., altitude, latitude, and longitude) of one-hop neighboring nodes to one-hop neighboring nodes to form a neighbor set comprised of two-hop neighboring nodes. During the route selection stage, the source node uses mobility metrics (M.1) (i.e., distance and velocity), the geographical location (M.3) of the destination node, and the link expiration time (M.2) between the source node and the destination node to select a route comprised of links with higher expected link expiration time to increase the route lifetime between nodes (i.e., source and destination). During the route maintenance stage, the link expires (or disconnects) and so the source node recalculates the link expiration time of another link in order to reestablish routes.

The routing scheme is compared with a state-of-the-art scheme that uses hello messages and topology control messages to discover and disseminate link state information. Random waypoint mobility model (A.2) is used in simulation. The routing scheme has been shown to increase QoS (P.1) (i.e., lower outage time, which contributes to lower packet loss rate) and to reduce energy consumption (P.2).

#### 5.2.3. Enhancing Routing Performance Using Probability of Link Disconnection

Ganbayar et al. [125] propose a proactive routing scheme to select routes based on the probability of link disconnection between nodes in a route as a result of high mobility in order to prolong route lifetime. The main objectives are to enhance route stability (O.1) as well as to enhance routing performance and QoS (O.3) in FANETs. The routing scheme addresses two challenges, namely high dynamicity (X.1) of network topology as well as low residual energy (X.3) due to retransmission. The routing scheme is designed for source routing (C.1.2).

During the neighbor and route discoveries stage, the source node periodically broadcasts RREQ messages that contain mobility metrics (i.e., nodes’ velocity) (M.1) and geographical location (M.3) of one-hop neighboring nodes to form a neighbor set comprised of two-hop neighboring nodes and to gather node information along potential routes. Intermediate nodes that receive RREQ calculates the link expiration time (M.2) and the probability of route breakage based on mobility of nodes. During the route selection stage, the destination nodes, which receive RREQ messages via different routes, select a route that provides higher link expiration time (i.e., higher connectivity time between source and destination) and lower probability of route breakage in order to prolong route lifetime and to reduce delay. The destination node sends a RREP message to the source node using the same route that the RREQ message traverses. During the route maintenance stage, link disconnection occurs due to the dynamicity of network topology and so the source node determines an alternative route before the expiration of the link expiration time.

The routing scheme is compared with a state-of-the-art scheme, which is a conventional routing scheme that establishes routes to destinations in an on-demand manner and supports both unicast and multicast routing. Random waypoint mobility model (A.2) is used in simulation. The routing scheme has been shown to increase route setup success rate (P.4) and to reduce energy consumption (P.2).

### 5.3. Reactive

This section presents two reactive routing schemes for FANETs.

#### 5.3.1. Enhancing Routing Performance in Vehicular Networks Using UAV-Assisted Approach

Omar et al. [101] propose a reactive routing scheme that enables UAVs to establish routes among themselves and with vehicular nodes on land when network partitions occur among the vehicular nodes. This means that, during normal operation, vehicular nodes communicate among themselves; however, when network partitions occur, some vehicular nodes are unable to communicate with each other. Therefore, the vehicular nodes establish routes with UAVs; specifically, a route consists of several parts, including links between vehicles, between vehicles and UAVs, and between UAVs. The main objectives are to enhance the route stability (O.1) as well as to enhance routing performance and QoS (O.3) in FANETs. The routing scheme addresses two challenges, namely high dynamicity (X.1) of network topology and cost (X.2), particularly routing overhead. The routing scheme is designed for source routing (C.1.2).

During the neighbor and route discoveries stage, each node broadcasts hello messages to form a neighbor set comprised of two-hop neighboring nodes. The hello message contains the geographical location (M.3), node identity (M.5), and one-hop neighboring nodes. Subsequently, the source node broadcasts RREQ messages that accumulate the link expiration time (M.2) along potential routes. Shorter distance between a pair of UAVs tends to increase the link expiration time and, hence, provides a longer link connectivity. During the route selection stage, the destination nodes receive RREQ messages via different routes and select a route with the highest link expiration time among the routes based on the minimum link expiration time of each route in order to prolong route lifetime. The destination node sends a RREP message to the source node using the same route that the RREQ message traverses.

The routing scheme is compared with a state-of-the-art scheme, which is a conventional routing scheme that establishes routes to destinations in an on-demand manner and supports both unicast and multicast routing. Random walk mobility model (A.1) is used in simulation. The routing scheme has been shown to increase QoS (P.1) (i.e., higher packet delivery rate and lower end-to-end delay) and route setup success rate (P.4).

#### 5.3.2. Enhancing Route Stability Using Link Delay

Biomo et al. [126] propose a reactive routing scheme, which is based on AODV [129], that enables UAVs to establish routes by accumulating a route stability factor, which is based on link delay and mobility. The main objectives are to enhance route stability (O.1) as well as to enhance routing performance and QoS (O.3) in FANETs. The routing scheme addresses two challenges, namely high dynamicity (X.1) of network topology and cost (X.2), particularly routing overhead which increase energy consumption and computational complexity. The routing scheme is designed for hop-by-hop (C.1.1) and source routing (C.1.2) approaches.

During the neighbor and route discoveries stage, each node periodically broadcasts hello messages to form a neighbor set comprised of two-hop neighboring nodes. The hello message contains the sending time, geographical location (M.3) (i.e., GPS coordinates), mobility metric (M.1) (i.e., the distance with and average speed of neighboring nodes), and one-hop neighboring nodes. Using the sending time information and the reception time at a neighboring node, the link delay can be estimated. Subsequently, the source node broadcasts RREQ messages to accumulate the route stability factor, which is based on the link delay and mobility metric along potential routes [130,131]. During the route selection stage, the destination nodes, which receive RREQ messages via different routes, select a route that provides the least accumulated route stability factor in order to increase route stability and to reduce delay. The destination node sends a RREP message to the source node using the same route that the RREQ message traverses. During the route maintenance stage when a link disconnection occurs, an intermediate node performs a local link repair so that the rest of the route can still be used and identifying a new route is not necessary. Specifically, the intermediate node does the following:stores the affected packets;broadcasts RREQ to identify a next-hop node located closer to the destination node, which is based on the greedy geographic forwarding (GGF) [132];selects the next-hop node to form a new link in order to repair the broken route;forwards packets along the new link. Nevertheless, if steps broadcasting RREQ and selecting the next-hop node are unsuccessful, then the node drops the packets in its storage.

The routing scheme is compared with a state-of-the-art scheme that prefers the route with the least number of hops during route discovery. Random waypoint mobility model (A.2) is used in simulation. The routing scheme has been shown to increase QoS (P.1) (i.e., higher packet delivery rate and lower routing overhead).

### 5.4. Hybrid

This section presents two hybrid routing schemes for FANETs.

#### 5.4.1. Enhancing Route Stability Using Energy-Efficient Technique

Ali et al. [102] propose a hybrid routing scheme that enables nodes to form clusters and routing by adjusting the transmission range of nodes in order to reduce energy consumption and computational complexity. The main objectives are to enhance route stability (O.1) and network coverage (O.2) in FANETs. The routing scheme addresses two challenges, namely high dynamicity (X.1) of network topology and low residual energy (X.3) as a result of long transmission range and packet retransmission. This routing scheme is designed for source routing (C.1.2).

During the neighbor and route discoveries stage, each node periodically broadcast messages (e.g., hello messages) to form a neighbor set comprised of two-hop neighboring nodes—the hello message contains its geographical location (M.3) and node identity (M.5)—as well as to predict the geographical locations of two-hop neighboring nodes based on coordinates, which indicates the distance between nodes. During the route selection stage, in order to increase network stability and hence the route lifetime, the underlying network is clustered using the K-mean algorithm based on a) residual energy (M.4) and mobility metrics (M.1), whereby nodes with higher residual energy and moving close to the average speed of the average speed are selected as CHs, and b) on geographical location (M.3) and node identity (M.5), whereby nodes moving close to the average speed of a CH joins its cluster. The CH collects data from CMs and then aggregates and forwards the data. The transmission power of CHs and CMs are adjusted to further reduce energy consumption. CHs are responsible for sending the data between the clusters. The route between CHs are established by considering the residual energy (M.4) of the CH. Hence, a CH with higher residual energy is selected as intermediate node. During the route maintenance stage, link disconnection occurs due to the dynamicity of network topology, and so, the route discovery mechanism is initiated to select a new CH or to join another CH as well as to reestablish routes.

The routing scheme is compared with a state-of-the-art scheme that uses the swarm-based algorithm to provide an optimized resource utilization. Random walk mobility model (A.1) is used in simulation. The routing scheme has been shown to reduce energy consumption (P.2) and the number of clusters in the network (P.3).

#### 5.4.2. Enhancing Routing Performance Using the Node Density of UAVs

Yu et al. [103] propose a hybrid routing scheme that enables nodes in a swarm of UAVs (or multi-UAV swarm) to form clusters and to establish a route in which nodes are geographically nearer to each other in order to increase link expiration time. The main objective is to enhance route stability (O.1) in FANETs. The routing scheme addresses the challenge of high dynamicity (X.1) of network topology. The routing scheme is designed for source routing (C.1.2).

During the neighbor and route discoveries stage, each node periodically broadcasts hello message to form a neighbor set comprised of two-hop neighboring nodes. The hello message contains the three-dimensional geographical location of nodes. During the route selection stage, in order to increase network stability and hence the route lifetime, the underlying network is clustered based on (a) residual energy (M.4) and mobility metrics (M.1), whereby nodes with higher residual energy and moving close to the average speed of the nodes in the neighbor set are selected as CHs, and (b) link expiration time (M.2) and geographical location (M.3), whereby nodes geographically close to the CH joins its cluster to reduce the latency of data dissemination and to improve network connectivity in order to increase route stability in both inter- or intra-cluster communication. During the route maintenance stage, link disconnection occurs due to the dynamicity of network topology, and so, the route discovery mechanism is initiated to select a new CH or to join another CH as well as to reestablish routes. The CH collects data from CMs and then aggregates and forwards the data. CHs are responsible for sending the data between the clusters as well. The route between CHs are established by considering the mobility metrics (M.1) (i.e., average distance), whereby nearest located CH is selected as an intermediate node to route the data. During the route maintenance stage, which is required when there are lower number of CMs in a cluster than a predefined threshold, the node initiates the route discovery mechanism (or the first stage) to re-cluster the network and to reestablish routes.

The routing scheme is compared with a state-of-the-art scheme, which is a cluster-based routing scheme. Realistic mobility model (A.6) is used in simulation. The routing scheme has been shown to increase QoS (P.1) as well as to reduce energy consumption (P.2) and the number of clusters in the network (P.3).

## 6. Open Issues

FANETs possess unique characteristics (e.g., high mobility degree, high dynamicity of network topology, low energy constraint, as well as different node densities in different network scenarios) that require novel routing schemes, although there are limited routing schemes for FANETs in the literature, but this survey paper can motivate researchers by identifying more research gaps in the respective domains. For instance, the high mobility degree characteristic has brought about three challenges to routing:high dynamicity (X.1), whereby nodes with high mobility can cause significant changes to network topology;high cost (X.2), whereby nodes with high mobility can cause link disconnections, resulting in high packet retransmission due to route failure, route discovery, and route maintenance;low residual energy (X.3), whereby higher number of message exchanges can cause high energy consumption and shorter route lifetime.

Novel routing schemes can cater to the unique characteristics in order to achieve better network performance, such as higher QoS performance (P.1) (e.g., higher throughput, as well as lower packet loss rate and delay) and lower energy consumption (P.2). In addition, novel routing schemes must be designed to cater for next-generation applications, which have stringent QoS requirements, such as high throughput as well as low latency and packet loss rate in real-time surveillance with video, voice, and images. Table 5 depicts the summary of open issues. The rest of this section presents open issues that can be pursued in this research topic.

### 6.1. Minimizing the Effects of Frequent Link Disconnections to Improve Routing

Frequent link disconnections and network partitions can increase route discovery and maintenance to reestablish routes as well as increase packet retransmission [133]. Consequently, network performance degrades, which can be unacceptable in critical circumstances, such as disaster relief and rescue operations [104,134]. There are two main challenges in improving network connectivity: a) high dynamicity (X.1) of network topology and b) high cost (X.2) due to increased routing overhead, energy consumption, and computational cost. Addressing this open issue helps to achieve higher QoS performance (P.1), lower energy consumption (P.2), and higher route setup success rate (P.4). Further investigation could be pursued to reduce link disconnections and network partitions; for instance, the next geographical location of a UAV can be predicted and used to maintain a route or to select intermediate nodes of a route.

### 6.2. Performing Routing in the Multi-UAV Swarm Scenarios

In the multi-UAV swarm scenarios, a massive amount of data is generated by a large collection of autonomous, small-sized, and lightweight (or non-weight-carrying) UAVs, which is a scenario called ultra-densification. Multi-UAV swarm also possess unique characteristics of FANETs, such as high mobility degree and high dynamicity of network topology. In addition to high dynamicity (X.1) of network topology, high cost (X.2), and low residual energy (X.3), there are two new challenges: (a) high channel demand requires high channel capacity [135,136], and (b) high collision rate among UAVs in a large swarm of UAVs requires reliable communication [137]. Further investigation could be pursued to investigate routing for reliable communication under different mobility models with different social behaviors of swarms (e.g., a school of fish, a swarm of bees, a swarm of ants, and a pack of wolves), levels of ultra-densification, and collision rates. Further investigation could also be pursued to design context-aware routing schemes that select next-hop and backhaul nodes (or BS) based on internal and external ambient factors, such as geographical location, the amount of fuel, and hardware capabilities, in order to establish long-term routes [93].

### 6.3. Performing Clustering for Supporting Routing in Multi-UAVs

Due to ultra-densification in the multi-UAVs and multi-UAV swarm scenarios, essential collaborative tasks, such as data aggregation, load distribution, resource allocation, and local synchronization, are performed to cater to the massive amount of data generated by UAVs. Clustering (i.e., hierarchical routing) segregates nodes with similar characteristics and behaviors into logical groups in order to improve network stability and scalability; hence, it is suitable to support the collaborative tasks [138]. Since clustering can increase network lifetime, it can increase route stability and lifetime [139]. While clustering schemes have been proposed for FANETs [103], there is a lack of focus on scenarios under ultra-densification. Since the network densification changes with geographical locations and time, further investigation could be pursued to perform clustering using context-aware approaches, such as artificial intelligence approaches and bio-inspired algorithms [140].

### 6.4. Enhancing Mobility Models for the Investigation of Routing in FANETs

The mobility pattern of the UAVs are highly dynamic in the three-dimensional space, and they may be spaced far apart, causing link disconnection that reduces route lifetime. In addition to high dynamicity (X.1) of network topology, high cost (X.2), and low residual energy (X.3), there are two new challenges: (a) high mobility degree that causes the next geographical locations of nodes to become unpredictable and (b) movement in the three-dimensional space, which differs from the two-dimensional space in MANETs and VANETs [137]. Nevertheless, some routing schemes have been investigated under two-dimensional space in FANETs, including those in References [102,103,112]. To the best of our knowledge, an extensive mobility model that covers the two new challenges are yet to be investigated to support the investigation of routing in FANETs [141]; nevertheless, movement with high mobility degree in a three-dimensional space is the core characteristic of FANETs. Further investigation could be pursued to enhance the mobility models so that the mobility characteristics and flight routes of UAVs emulate real-life scenarios in FANETs [142].

### 6.5. Improving Network Performance and Survivability through Multi-Pathing

Frequent link disconnections and network partitions occur due to high dynamicity (X.1) of network topology, and so, route discovery and maintenance must be performed frequently to reestablish routes, resulting in higher cost (X.2). Multi-pathing establishes more than a single route so that (a) multiple routes can be used at the same time to maximize resource utilization [143,144] and to reduce network congestion [145,146], which is important under ultra-densification, and (b) a backup route is available when the current route is broken to improve network survivability and fault tolerance [147], contributing to reduced routing overhead [148]. Further investigation could be pursued to investigate multi-pathing to improve network performance and survivability. Context-aware approaches, such as artificial intelligence approaches and bio-inspired algorithms [140], can be adopted to cater to the dynamic network condition that affects multi-pathing.

### 6.6. Improving Network Performance by Using Artificial Intelligence

Artificial intelligence (AI), such as deep learning and reinforcement learning, allows a system to autonomously learn from its operating environment and past experience in order to improve its performance as time goes by. AI plays a vital role in optimizing communication among flying nodes [139,149]. For instance, reinforcement learning enables UAVs to learn from their respective operating environment and to adjust their position and movement (i.e., speed and direction) accordingly. Learning enables UAVs to self-organize and to select optimal routes among themselves. Furthermore, AI can be embedded in ground stations to enhance FANETs, such as to detect the presence of UAVs, to monitor their movement and behavior as time goes by, to create channel model and utilization map, and to distribute traffic load among routes in the multi-UAV swarm scenario. Further investigation could be pursued to explore and exploit the AI domain to improve routing performance in FANETs.

### 6.7. Improving Network Coverage by Using High and Low Altitude UAVs

Frequent link disconnections and network partitions occur due to high dynamicity (X.1) of network topology, particularly when nodes move out of the coverage (or transmission range) of neighboring nodes or BS. This increases route discovery and maintenance to reestablish routes, as well as packet retransmission, resulting in higher cost (X.2) [133]. Consequently, network performance degrades, which can be unacceptable in critical circumstances, such as during disaster relief and rescue operations [104,134]. Both high and low altitude platforms can be used. The high altitude platforms (e.g., balloons) provide a large coverage that can improve connectivity among nodes [150,151], hence improving the routing performance. The low altitude platforms (e.g., helikites [152]) provide a small coverage that can improve local connectivity among nodes [153]. Further investigation could be pursued to investigate how UAVs can collaborate with high and low altitude platforms to improve routing performance.

### 6.8. Reducing Energy Consumption Using Green Energy

UAVs are highly dynamic (X.1) in the three-dimensional space, and they may be spaced far apart, causing frequent link disconnection and network partitions that reduce route lifetime. Hence, high energy consumption is incurred to establish and select routes from time to time. Green energy-based mechanisms have been introduced to reduce energy consumption [26]. It helps to keep the energy by reducing the number of exchange messages between UAVs. Further investigation could be pursued to investigate how UAVs can collaborate for energy harvesting. It provides the longer life of network and route by keeping UAVs alive. Solar panels can be used to harvest energy in order to minimize the drainage of power in UAVs. Addressing this open issue helps to achieve lower energy consumption (P.2) and high route setup rate (P.4).

## 7. Conclusions

This paper presents a review of the limited work on routing in flying ad hoc networks (FANETs), which possess unique characteristics (e.g., movement with high mobility degree in the three-dimensional space) that distinguish itself from the traditional mobile ad hoc networks and vehicular ad hoc networks. Routing in FANETs aims to achieve the objectives of enhancing route stability, network coverage, as well as routing performance and quality of service while addressing the challenges of high dynamicity, high cost, and low residual energy. Various routing metrics, such as mobility metrics, link expiration time, geographical location, and residual energy, are used to select routes; for instance, routes with higher link expiration time are selected. Due to the limited research done on routing in FANETs as well as the limited focus on the unique characteristics of FANETs, there remains a large amount of future work, and this paper has laid a solid foundation to motivate investigations of the open issues raised in this topic.

## Figures and Tables

**Figure 1 sensors-20-00038-f001:**
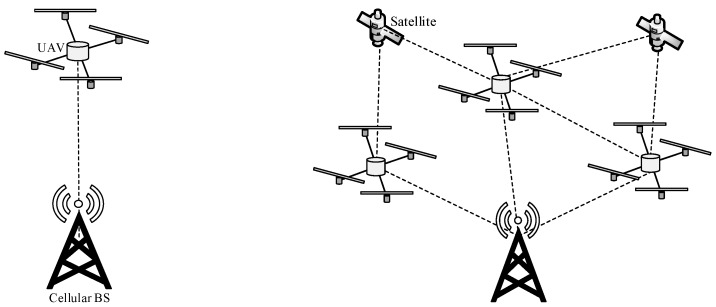
Two main scenarios in flying ad hoc networks (FANETs).

**Figure 2 sensors-20-00038-f002:**
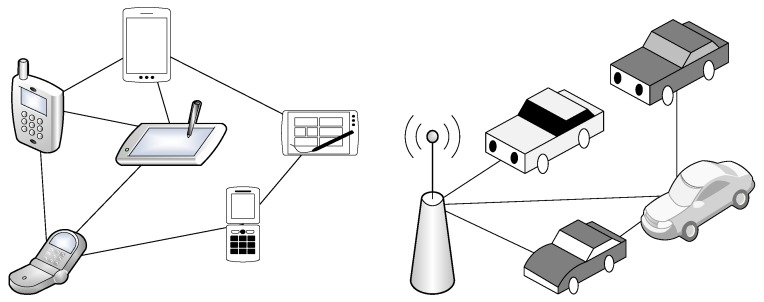
Two categories of ad hoc networks, namely mobile ad hoc networks (MANETs) and vehicular ad hoc networks (VANETs): The solid line represents the connectivity between two nodes.

**Figure 3 sensors-20-00038-f003:**
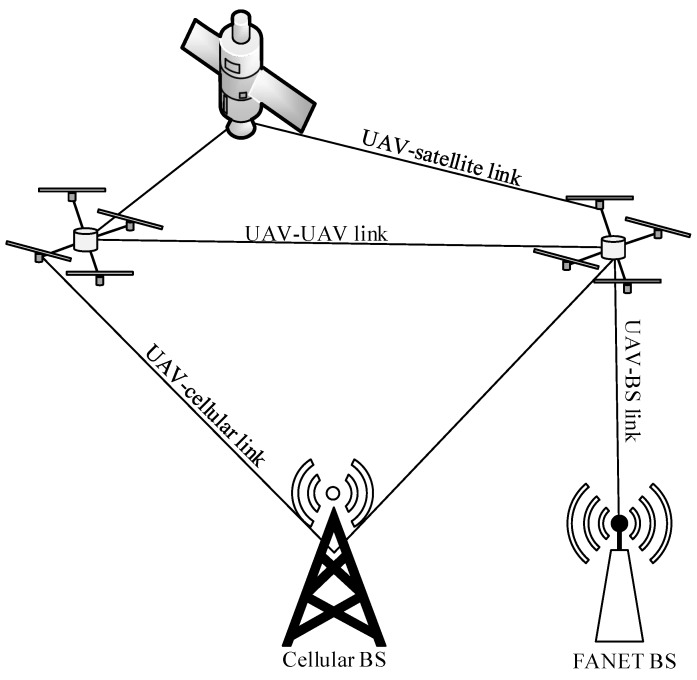
Four types of links in FANETs.

**Figure 4 sensors-20-00038-f004:**
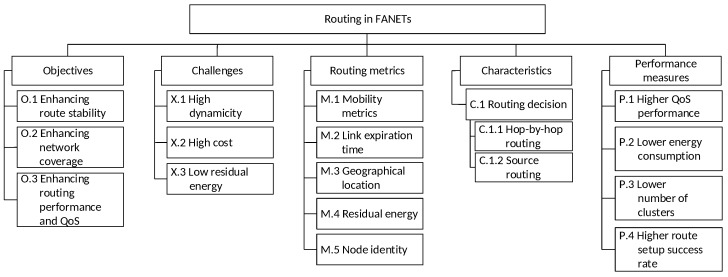
A taxonomy of routing attributes in FANETs.

**Table 1 sensors-20-00038-t001:** Comparison of existing survey papers in FANETs with our paper.

Reference	Year	Topic			Focus											
		Routing	Mobility Models	Applications	Motivation	Comparison with other ad hoc Networks	Requirements	Mobility Models	Taxonomy	Objectives	Challenges	Routing Metrics	Characteristics	Performance Measures	Frame work	Open issues
Axel et al. [2]	2010			×			×									×
Bauer et al. [28]	2011		×				×						×			×
Neji et al. [29]	2013			×			×				×		×	×		
Bekmezci et al. [7]	2013			×	×	×	×	×			×		×	×		×
Ozgur et al. [18]	2013	×				×					×		×	×		×
Xie et al. [30]	2014		×			×		×			×			×		×
Naser et al. [31]	2016			×		×				×	×		×	×		×
Gupta et al. [32]	2016	×			×	×					×	×	×	×		×
Samira et al. [33]	2016			×			×		×	×	×					×
Armir et al. [34]	2017		×		×		×	×						×		×
Omar et al. [35]	2017	×					×	×	×		×		×	×		×
Zeeshan et al. [36]	2018			×	×		×				×			×		×
Khan et al. [37]	2018	×				×		×	×		×		×	×		
Antonio et al. [38]	2018		×			×		×					×	×		×
Kaur et al. [39]	2018	×			×	×					×	×	×	×		
Otto et al. [40]	2018			×			×			×	×	×	×	×		×
Jinfang et al. [41]	2018	×				×			×				×	×		×
Our paper	2019	×			×	×	×	×	×	×	×	×	×	×	×	×

A cross × indicates that the option, which is represented by the column, applies to the reference, which is represented by the row.

**Table 2 sensors-20-00038-t002:** Comparison of FANETs with MANETs and VANETs.

Category		MANETs	VANETs	FANETs
Types of link	Ad hoc	Yes	Yes	Yes
	Direct link	Yes	Yes	Yes
	Satellite	No	No	Yes
	Cellular	No	Yes	Yes
Characteristics	Mobility degree	Low	Medium	High
	Mobility Models	Random way point	Prediction based	SRCM, Realistic
	Energy constraint	High	Low	Medium
	Radio propagation model	NLOS	NLOS	LOS
	Localization method	GPS	Assisted-GPS, differential-GPS	Inertia measurement unit
	Node density	High	Medium	Low

**Table 3 sensors-20-00038-t003:** Stages of routing frameworks in FANETs.

Category	Stage	Details	Outcomes
Adaptive	First	Nodes exchange messages among themselves and prediction to localize the nodes in the space	Neighbor sets, network typologies, and location
	Second	Use routing metrics and select route	Establishment of routing path
Proactive	First	Nodes exchange messages among themselves	Neighbor sets and network typologies are formed
	Second	Use routing metrics and select route	Establishment of routing path
	Third	Reestablishment of route to cater dynamicity	New routes are established
Reactive	First	Nodes exchange messages among themselves and send route requests (RREQs) from source nodes towards the destination node	Neighbor sets, network typologies, and route identification
	Second	Response of RREQ from destination node towards the source node	Route chosen by RREP
Hybrid	First	Nodes exchange messages among themselves	Neighbor sets and network typologies are formed
	Second	Non-clustered nodes elect CHs	CHs are elected
	Third	Non-clustered nodes join clusters	Clusters are formed

**Table 4 sensors-20-00038-t004:** Summary of objectives, metrics, and performance of routing schemes proposed in the literature for FANETs.

Reference	Year	Approach				Objectives			Challenges			Routing Metrics					Charac-Teristic		Performance Measures			
		F.1 Adaptive	F.2 Proactive	F.3 Reactive	F.4 Hybrid	O.1 Enhancing routing stability	O.2 Enhancing network coverage	O.3 Enhancing routing performance and QoS	X.1 High dynamicity	X.2 High cost	X.3 Low residual energy	M.1 Mobility metrics	M.2 Link expiration time	M.3 Geographical location	M.4 Residual energy	M.5 Node identity	C.1.1 Hop-by-hop routing	C.1.2 Source routing	P.1 Higher QoS performance	P.2 Lower energy consumption	P.3 Lower number of clusters	P.4 Higher route setup success rate
Zheng et al. [100]	2018	×				×		×	×	×				×		×	×		×			×
Khelifi et al. [99]	2018	×						×	×	×	×	×			×		×	×	×	×		
Alshabtat et al. [112]	2010		×				×	×	×	×		×					×		×			
Rosati et al. [124]	2016		×			×		×	×			×	×	×				×	×	×		
Ganbayar et al. [125]	2017		×					×	×		×	×			×			×		×		×
Omar et al. [101]	2017			×		×		×	×	×			×	×		×		×				
Biomo et al. [126]	2014			×		×	×		×	×		×		×			×	×	×			
Ali et al. [102]	2018				×	×	×				×	×		×	×	×		×		×	×	
Yu et al. [103]	2016				×	×			×			×	×	×	×			×	×	×	×	

**Table 5 sensors-20-00038-t005:** A Summary of open issues and their purposes, challenges, and proposed solutions.

Open Issue	Purpose	Challenges	Proposed Solutions
Minimizing the effects of frequent link disconnections to improve routing	Reducing packet retransmission and reestablishing routes	High dynamicity High cost	Predicting the next geographical location of a UAV in route selection and maintenance.
Performing routing in the multi-UAV swarm scenarios	Managing massive amount of data due to ultra-densification	High dynamicity High cost Low residual energy	Predicting the next geographical location of a UAV.
Performing clustering for supporting routing in multi-UAVs	Deploying collaborative tasks, including data aggregation, load distribution, and resource distribution	High cost Low residual energy	Using context-aware approaches, such as artificial intelligence approaches and bio-inspired algorithms
Enhancing mobility models for the investigation of routing in FANETs	Mobility management	High dynamicity High cost Low residual energy	Forming mobility models based on real-life scenarios
Improving network performance and survivability through multi-pathing	Maximizing resource utilization and reducing network congestion	High routing overhead	Artificial intelligence (AI)-based approaches
Improving network performance by using artificial intelligence	Optimize performance	High cost High routing overhead	Using AI approaches to improve network performance
Improving network coverage by using high and low altitude unmanned aerial vehicles (UAVs)	Reducing packet retransmission and reestablishing routes	High cost	Enabling collaboration between high and low altitude platforms
Reducing power consumption by using green energy	Reducing network partitioning	High cost Low residual energy	Use solar panels for extra energy backup

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
