# Peer review of "Routing Schemes in FANETs: A Survey"

_sensors, 2019, doi:10.3390/s20010038_

Round 1
Reviewer 1 Report
Please find below my comments/suggestions
Add footnote for table 1 to specify what is X. If everything is already discussed in past recent works, what is new in this survey? I am not fully convinced from the contributions of paper.
Remove paper structure Figure 2.
Table 2 is referred on page 5 and it is on page 8. It should be on the same or next page.
Add machine learning based papers related to the topic.
Add Open Issues figure to summarize the research directions and recommendations.
Reviewer 2 Report
The authors present a paper on a current topic for communications networks: FANET (Flying adhoc network). The authors propose a review of routing techniques in this type of networks. Finally, the authors address some problems that should be improved even in these types of networks (open issues). The text is well written and structured, easy to read and presents a set of references that are updated and coherent with the length of the paper.
The paper can be improved by including several aspects as described below.
1. The abstract exceeds 200 words (see instructions for authors).
2. In the introduction, on lines 50-59, a structure of the itemize type can be used to improve the understanding and visibility of the paragraph. The same happens in other parts of the text.
3. Table 1 appears before being cited.
4. Thanking the authors about Figure 2 greatly clarifies what they want to reflect on the paper.
5. Figure 3 could be removed if there were space limitations.
6. The review of mobility models in section 2.1.2 is very simple (lines 152-157). For example, the authors seem to indicate that in the MANETs only the 'randow waypoint model'(RWP) is used. This mobility model presents many problems and was replaced by others as can be studied in the literature:
Yoon, J., Liu, M., & Noble, B. (2003, March). Random waypoint considered harmful. In IEEE INFOCOM 2003. Twenty-second Annual Joint Conference of the IEEE Computer and Communications Societies (IEEE Cat. No. 03CH37428) (Vol. 2, pp. 1312-1321). IEEE.
Although it is true that RWP has been used a lot, you could mention someone else, for example, Random trip model:
Le Boudec, J. Y., & Vojnovic, M. (2006). The random trip model: stability, stationary regime, and perfect simulation. Ieee/Acm Transactions On Networking, 14(6), 1153-1166.
7. In line 159, it is said that the energy restrictions of the FANETs are moderate, could any reference be included?
8. Table 2 could approach the text where it is cited.
9. Line 296. Missing parentheses
10. Section 5, the most important of the paper, reviews the routing algorithms. The different subsections are very interesting and provide an extra to the review. It would be advisable to include some more paper this year or recent, especially in the adaptive section.
11. The explanation of each routing algorithm always ends with its advantages, for example, the packet delivery ratio. But it is not indicated with what other algorithm has been compared to establish that improvement.
12. The last section on open problems could be expanded with other aspects such as: green energy or load balancing in the network.
Reviewer 3 Report
This paper presents a survey for routing schemes in aerial ad-hoc networks. The topic is quite timely, while the organization of the paper is clear. The authors also provide open issues for aerial networks, something that adds value in survey papers. However, there are also some important issues to be addressed:
1) The presentation of the paper needs significant improvement. There are several typos and grammar errors, while articles are missing in many cases. The authors should thoroughly proofread the paper or refer to a professional editing service. Some indicative mistakes are as follows:
"has *an extensive range" "plays *a significant role" "in *the three-dimensional" "the mobility pattern...*is highly" "establishing long-term....*is essential" "flying nodes naturally *have" "this paper is *the first"and many others. Please note that the aforementioned errors are only indicative.
2) The authors should add recent works on UAVs to highlight the timeliness of the topic. Please see:
P. Mekikis et al., "Communication recovery with emergency aerial networks," in IEEE Transactions on Consumer Electronics, vol. 63, no. 3, pp. 291-299, August 2017. J. Ye et al., "Secure UAV-to-UAV Systems With Spatially Random UAVs," in IEEE Wireless Communications Letters, vol. 8, no. 2, pp. 564-567, April 2019. P. Mekikis and A. Antonopoulos, "Breaking the Boundaries of Aerial Networks with Charging Stations," IEEE International Conference on Communications (ICC), Shanghai, China, 2019, pp. 1-6. S. Zhang et al., "Cellular UAV-to-X Communications: Design and Optimization for Multi-UAV Networks," in IEEE Transactions on Wireless Communications, vol. 18, no. 2, pp. 1346-1359, Feb. 2019.3) In the same context, the authors should make a thorough review on the topic of their work to include all the relevant recent studies, e.g.:
I. Mahmud and Y. Cho, "Adaptive Hello Interval in FANET Routing Protocols for Green UAVs," in IEEE Access, vol. 7, pp. 63004-63015, 2019. M. Y. Arafat and S. Moh, "Location-Aided Delay Tolerant Routing Protocol in UAV Networks for Post-Disaster Operation," in IEEE Access, vol. 6, pp. 59891-59906, 2018.4) The authors claim several times that the work on this topic is limited. However, this limits the contribution of this article. What is the point of presenting a survey if the work in a given topic is limited? If the authors want to motivate more research in this topic, this should be written clearly and consistently throughout the paper.
Reviewer 4 Report
In this paper authors provide a deep overview about routing schemes in FANETs. The paper is well structured and it is quite easy to follow.
Although the paper itself does not provide any routing proposal, it is a good starting point for researchers that begin in FANEts research.
In my opinion, this is a good overview paper about FANETS taxonomy.
Round 2
Reviewer 1 Report
I am satisfied with the revised manuscript.
Reviewer 2 Report
The authors have made all the changes that had been proposed. The paper has been sufficiently improved with the requests of all reviewers.
Reviewer 3 Report
The authors have addressed my comments.